# The Effect of Chrysin-Loaded Phytosomes on Insulin Resistance and Blood Sugar Control in Type 2 Diabetic db/db Mice

**DOI:** 10.3390/molecules25235503

**Published:** 2020-11-24

**Authors:** Seong-min Kim, Jee-Young Imm

**Affiliations:** Department of Foods and Nutrition, Kookmin University, Seoul 02707, Korea; ksm618@nate.com

**Keywords:** chrysin, phytosome, blood sugar control, C57BLKS/J-db/db mice, insulin resistance

## Abstract

Although a variety of beneficial health effects of natural flavonoids, including chrysin, has been suggested, poor solubility and bioavailability limit their practical use. As a promising delivery system, chrysin-loaded phytosomes (CPs) were prepared using egg phospholipid (EPL) at a 1:3 molar ratio and its antidiabetic effects were assessed in db/db diabetic mice. Male C57BLKS/J-db/db mice were fed a normal diet (control), chrysin diet (100 mg chrysin/kg), CP diet (100 mg chrysin equivalent/kg), metformin diet (200 mg/kg) or EPL diet (vehicle, the same amount of EPL used for CP preparation) for 9 weeks. Administration of CP significantly decreased fasting blood glucose and insulin levels in db/db mice compared with the control. An oral glucose tolerance test and homeostatic model assessment for insulin resistance were significantly improved in the CP group (*p* < 0.05). CP treatment suppressed gluconeogenesis via downregulation of phosphoenolpyruvate carboxykinase while it promoted glucose uptake in the skeletal muscle and liver of db/db mice (*p* < 0.05). The CP-mediated improved glucose utilization in the muscle was confirmed by upregulation of glucose transporter type 4, hexokinase2 and peroxisome proliferator-activated receptor γ during treatment (*p* < 0.05). The CP-induced promotion of GLUT4 plasma translocation was confirmed in the skeletal muscle of db/db mice (*p* < 0.05). Based on the results, CP showed greater antidiabetic performance compared to the control by ameliorating insulin resistance in db/db mice and phytosome can be used as an effective antidiabetic agent.

## 1. Introduction

Diabetes mellitus (DM) is a metabolic disorder characterized by chronic hyperglycemia. Impairment of insulin secretion, increased glucose hepatic output, and decreased insulin sensitivity are responsible for DM [1]. Type 1 DM is caused by destruction of the insulin producing pancreatic β-cell islet while insulin resistance is the main cause for type 2 DM. Most diabetic patients are type 2 DM and it is becoming an emerging global health concern [2]. Genetic predisposition, obesity, and lack of exercise are major risk factors for type 2 DM. The pancreatic β-cell increases insulin production for glucose disposal in the early stage of insulin resistance; however, extended hyperglycemic conditions eventually lead to type 2 DM [3].

Flavonoids are plant metabolites that are abundantly found in the human diet. Consumption of flavonoids, especially anthocyanins, lowered risk of type 2 DM in the US population [4] and the consumption of flavone and flavonols was inversely correlated with insulin resistance in the 2007–2009 Korea National Health and Nutritional Examination surveys [5]. Chemically, flavonoids consist of two aromatic rings (A and B) linked by an oxygenated three carbon heterocyclic ring (C). The number and location of hydroxylation, methoxylation and glycosylation in flavonoids critically influences their bioactivity [6].

Chrysin (5,7-dihydroxyflavone) is a natural flavonoid distributed in plants and often found in bee propolis and flowers. Chrysin treatment for 16 weeks significantly reduced renal inflammation by the suppression of renal TNF-α production and NF-κB nuclear translocation in high-fat diet and streptozotocin-induced type 2 DM rats. The reduction in inflammation and oxidative stress due to chrysin treatment resulted in the restoration of renal function [7]. The administration of chrysin (30 and 100 mg/kg for 26 days) in Streptozotocin-induced diabetic rats resulted in decreased blood glucose levels and inflammatory cytokine production in the cerebral cortex and hippocampus [8]. Chrysin treatment also ameliorated STZ-induced hypertension and vascular complications [9]. Chrysin administration (100 mg/kg, 18 weeks) was able to attenuate fructose-induced metabolic syndrome, such as hypertension, hepatic fibrosis, and hyperinsulinemia, but serum leptin and hepatic TG levels remained unchanged [10]. In addition, it has been reported that chrysin exerts neuroprotective effects by increasing the dopamine levels in the brain in 6-hydroxydopamine-induced neurodegenerative Parkinson’s disease mice [11].

The effect of dietary flavonoids is still not conclusive even though the evidence regarding health benefits of flavonoid intake is increasing [12]. This is partly due to the low absorption and fast clearance of flavonoids in the body. Most orally administered chrysin (400 mg) was excreted in feces and only a minor portion was detected in urine as chrysin glucuronide (2–26 mg) or chrysin (0.2–3.1 mg), based on the result of a bioavailability study with seven healthy volunteers [13]. Regarding the effect of chrysin on the intestinal environment and microbiome, the administration of chrysin (100 mg/kg, 18 weeks) decreased high fructose-mediated upregulated GLUT5 mRNA expression but did not improve dysbiosis in rats [14]. Poor solubility and short circulation half-life lowered the bioavailability of chrysin. Thus, an efficient delivery system is required to enhance the low bioavailability of chrysin.

Phytosomes refers to the molecular complexation between phytochemical constituents and phospholipids [15]. Typically, polyphenolic compounds are present as an integral part of phospholipid membrane in phytosomes and display better pharmacological performance compared to liposomes containing polyphenolic constituents in their inner cavity [16]. Rani et al. [17] has reported that the phytosome of Casuarina equisetifolia extract containing chrysin improved antidiabetic and antihyperlipidemic activities in Wistar rats.

In our previous study, stable (zeta potential −31 mV; polydispersity index 0.29) and uniform nano-sized chrysin-loaded phytosomes (CPs) with an average particle size of 117 nm were produced using egg phospholipid (EPL). We confirmed that CP promoted glucose uptake by upregulating GLUT4 and PPAR γ gene expression in C2C12 myotubes [18]. As a continuation, the previous study effects of CP on insulin resistance and blood sugar control were examined in type 2 diabetic db/db mice.

## 2. Results and Discussion

### 2.1. Effect of CP Supplementation on Blood Biochemical Makers

C57BL/KsJ-db/db mice have been widely used as a model of DM and this strain displays increased leptin and insulin level as well as high plasma cholesterol [19]. In our 9 week feeding trial, the animals were randomly divided into six groups: (1) m+/db, (2) db/db control group (control), (3) positive control group (metformin), (4) db/db + chrysin group (chrysin), (5) db/db + CP (CP), and (6) db/db + vehicle group (vehicle). No significant difference was found in initial and final body weight among all db/db groups after completing feeding trial (data not shown). The blood biochemical makers of each experimental group are listed in Table 1. The level of alanine aminotransferase (ALT), the liver inflammation marker, tends to decrease in the CP group (0.1 < *p* < 0.05). ALT has greater specificity for detecting liver injury than aspartate aminotransferase (AST) since ALT is mainly localized in the hepatocytes while AST is present in various tissues such as the liver, muscles, and kidneys [20]. Sirovina et al. [21] reported that the intraperitoneal injection of chrysin (50 mg/kg, 7 days) in alloxan-induced diabetic mice alleviated liver damage; however, no indication of the hepatoprotective effects of chrysin were found in this study.

Serum total cholesterol (TC) and low-density lipoprotein cholesterol (LDL-c) concentrations were significantly decreased in the CP group (*p* < 0.05). In addition, Triacylglycerol (TG), TC and LDLc levels were reduced in the vehicle group. The latter observation is in line with the observation by Lee et al. [22], who reported that administration of phosphatidylcholine reduced TG and TC levels in a high fat diet-induced obesity mice model. EPL consists of phosphatidylcholine (84.5%), phosphatidyl-ethanolamine (9%), and sphingomyelin (3%). Dietary PC also significantly lowered intestinal cholesterol absorption in rat and egg PC was more effective than soy PC [23]. The greater proportion of long-chain saturated fatty acid in egg PC could be associated with greater potency. The beneficial effect of dietary sphingomyelin either from milk or egg on lipid metabolism was reported in high-fat diet-fed mice [24].

No significant differences were found in serum leptin levels among experimental groups even though leptin is known to be related to body fat mass and insulin resistance [25]. Dietary chrysin supplementation did not decrease fat mass, serum leptin, or insulin levels in azoxymethane-induced colon carcinogenic mice [19]. The level of glycated hemoglobin (HbA1c) in the metformin group decreased significantly (*p* < 0.05). HbA1c reflects the extent of erythrocyte glycation accumulated over 2–3 months. Thus, HbA1c has been used for the evaluation of long-term glycemic control [26]. Considering that HbA1c reflects the interaction between protein and glucose, it can probably predict the risk of diabetic complications better than other glycemic control makers such as fasting plasma glucose and oral glucose tolerance test (OGTT) [27]. We are not sure about the exact reasons for the lack of statistical significance in HbA1c between control and CP group, but it may be ascribed to insufficient dosage or high biological variation in HbA1c.

### 2.2. Effect of CP on Insulin Resistance

The fasting blood glucose level (FBG) of the control group increased by five times with the m+/db group after 9 weeks. The administration of metformin and CP significantly lowered FBG (Figure 1; *p* < 0.05) but no change was found in chrysin group. All treatment groups displayed significant differences in serum insulin levels. Homeostasis model assessment of insulin resistance (HOMA-IR) was calculated to evaluate samples contribution to insulin resistance. HOMA-IR improvement was found in the metformin, chrysin and CP groups compared to the control group (*p* < 0.05). This result indicates that insulin sensitivity was effectively improved by CP rather than by chrysin supplementation.

The OGTT is widely used to evaluate an animal’s ability to control blood glucose. The rate of blood glucose disappearance was accelerated when metformin or CP was administrated to db/db mice (Figure 2A) and glucose tolerance expressed as the area under the curve (AUC) was significantly improved in the metformin and CP groups (Figure 2B, *p* < 0.05). There are no clear-cut criteria for diagnosing insulin resistance and some discrepancy was noticed in the biomarkers for insulin resistance such as FBG, HOMA-IR, and HbA1c. The metformin group exhibited consistent improvements in all three biomarkers, while the CP group displayed significant difference in both FBG and HOMA-IR. Chrysin treatment failed to show significant differences in FBG and HbA1c. Kang et al. [28] reported that FBG is critically influenced by stressful conditions and food intake while the rate of hemoglobin glycation varied depending on erythrocyte lifespan, iron status, and renal function.

These results suggest that chrysin bioavailability was increased by the complexation with EPL. Similarly, another study found that silybin, an active flavonoid in milk thistle, increased over 100-fold in plasma when administered as a nanophytosome. Furthermore, pharmacodynamic analysis of rats showed that silybin plasma concentrations increased 1.6 times when administered as a phytosome compared with a free silybin mixture [29]. Orally administered quercetin phytosome also improved plasma quercetin absorption in healthy volunteers (18−50 years) up to 20 times [30].

### 2.3. Effects of CP on the Glucose Metabolism Related Gene Expression in the Liver

Insulin resistance and increased hepatic glucose production lead to hyperglycemia in type 2 DM [31]. Gluconeogenesis is considered to be an important parameter of glucose homeostasis and the glucogenic pathway is suppressed by insulin via transcriptional regulation of the gluconeogenesis enzymes, glucose-6-phosphatase (G6Pase) phosphoenolpyruvate carboxykinase (PEPCK) [4,32]. Type 2 DM patients typically display increased hepatic gluconeogenesis. Metformin, one of the most effective drugs in treating type 2 DM, significantly reduces hepatic gluconeogenesis without increasing insulin secretion [33]. Thus, the gene expressions of two gluconeogenesis rate-limiting enzymes, G6Pase and PEPCK, are effective molecular targets for treating type 2 DM [32].

Gene expressions of G6Pase and PEPCK were significantly increased in db/db mice, reflecting increased glucose output in the liver as shown in Figure 3. The gene expressions of these two enzymes were significantly decreased through metformin treatment (*p* < 0.05) while CP treatment significantly suppressed only PEPCK expression alone (*p* < 0.05).

The hepatic HK2 level was significantly decreased in db/db mice compared with the m+/db, but it was significantly upregulated by metformin or CP treatment (*p* < 0.01, *p* < 0.05, Figure 3C). Li et al. [34] reported that all HK isoform (1–4) expressions were significantly reduced in the liver of db/db mice. Berberine treatment alleviated hyperglycemia by upregulating all hepatic hexokinase isoforms. We measured HK2 levels since HK2 has greater glucose affinity than other hexokinase isoforms [35]. Hepatic Hk2 gene plays an important role in glucose homeostasis and overexpressed HK2 expression in the liver suggested as an effective pharmacological target for type 2 DM [36]. These results indicate that the hypoglycemic effect of CP was partly due to suppression of PEPCK and upregulated hepatic HK2 in db/db mice.

PEPCK modulates gluconeogenic flux to glucose-6-phosphate by controlling oxaloacetic acid conversion to phosphoenolpyruvate, while G6Pase regulates the production of free glucose from glucose-6-phosphate [37]. Insulin resistance impairs effective action of insulin on glucogenic enzymes and results in hyperglycemia in type 2 DM. Liver-specific insulin-receptor knockout mice (LIRKO) displayed severe glucose intolerance and resulted in increased hepatic glucose production. LIRKO mice showed significantly increased PEPCK and G6Pase expressions in the liver accompanying hyperinsulinemia [38].

The supplementation of citrus unshiu peel extract ameliorated hyperglycemia by modulation of hepatic gluconeogenesis in db/db mice, similar to the results of this study [39]. Unshiu peel extract significantly suppressed PEPCK mRNA expression without affecting G6Pase and glucokinase expression in the liver. Gluconeogenesis suppression is suggested as a therapeutic target to alleviate type 2 DM Metformin, a positive control used in this study, known as a potent inhibitor of hepatic glucose production [32]. Downregulation of PEPCK by CP contributed to the regulation of glucose homeostasis by modulating the gluconeogenic flux.

### 2.4. Effects of CP on Glucose Uptake Related Gene Expression in the Skeletal Muscle

Changes in mRNA expression of GLUT4, HK2 and PPARγ in femoral muscle by sample treatment were analyzed using a qRT-PCR. GLUT4 gene expression was significantly upregulated by metformin and CP treatment (*p* < 0.05, Figure 4A).

HK2 gene expression increased by 1.5-fold in the CP group compared to the control group (*p* < 0.05, Figure 4B). PPARγ gene expression was significantly increased in the metformin, CP, and vehicle groups (*p* < 0.05, Figure 4C). Jeong et al. [40] reported that ginsenoside Rh4, Rg5, and Rk1 in black ginseng extract significantly upregulated PPARγ gene expression in muscle and improved insulin resistance in db/db mice. GLUT4 is a dominant insulin responsive glucose transporter in skeletal muscle. It promotes the active transport of glucose across the plasma membrane and HK2 contributed to glucose disposal of skeletal muscle by maintaining the glucose concentration gradient within the cell. Epinephrine and insulin transcriptionally regulated these glucose utilization steps in skeletal muscles [41]. Reduced GLUT4 expression and impaired insulin signaling lead to incomplete glucose tolerance (IGT) and hyperglycemia in type 2 DM [42]. Downregulated GLUT4 mRNA and protein expression in skeletal muscle are the main possible reasons for insulin resistance and is often observed in db/db mice [43,44]. GLUT4 overexpression in the skeletal muscle of db/db mice restored plasma glucose disposal and improved insulin sensitivity [45].

Oral supplementation of chrysin (50 mg/kg) resulted in an antidiabetic effect in diabetic athymic nude mice induced by nicotinamide injection [46]. However, a similar effect was not seen in this investigation. OGTT and changes in serum inflammatory cytokine production were analyzed under acute and subacute diabetic conditions (50 mg/kg, 10 days) using athymic nude diabetic mice in the previous study [46]. Unfortunately, they did not provide detailed information regarding the mechanisms of antidiabetic effect. Different animal models and experimental periods might affect the results even though the exact reasons for this discrepancy are not clear.

PPARγ exerts an insulin sensitizing effect and dysfunction of PPARγ causes insulin resistance and DM in humans [47]. PPARγ activation through ligand binding of thiazolidinediones enhanced insulin sensitivity in type 2 diabetic patients [48]. Furthermore, PPARγ-agonist also stimulated insulin-mediated suppression of gluconeogenesis in the liver [49]. In our previous investigation, CP significantly increased glucose uptake in C2C12 myotubes by activating PPARγ and GLUT4 [18]. CP-mediated glucose uptake promoting the effect observed in skeletal muscle cells was confirmed in db/db mice. CP promoted glucose utilization in skeletal muscles by activating glucose uptake-related gene expression in the skeletal muscle.

### 2.5. Effects of CP on GLUT4 Plasma Translocation in db/db Mice

GLUT4 normally located in the cytoplasm as vesicles in the skeletal muscle and translocated to plasma membrane in response to insulin or AMPK stimulation. Insulin resistance in skeletal reduced plasma GLUT4 translocation and glycogen synthesis in type 2 DM [50]. In order to confirm CP treatment leading to improved GLUT4 translocation, immunoblotting of plasma membrane GLUT4 was carried out. Plasma translocations of GLUT4 were increased by 1.6-fold in the CP group in the skeletal muscle of db/db mice compared to the control group (*p* < 0.05, Figure 5). This result indicates that the CP-induced improvement of insulin resistance in the db/db mice was probably caused by increased GLUT 4 translocation and insulin sensitivity in the skeletal muscle.

## 3. Materials and Methods

### 3.1. Materials

Egg phospholipid (EPL) was kindly provided by Doosan Corporation (Seoul, Korea). Chrysin and all other reagents were purchased from Sigma-Aldrich Inc. (St. Louis, MO, USA). The hemoglobin A1c (HbA1c), insulin, and leptin quantification kits were purchased from Crystal Chem (Downer Grove, IL, USA). Triacylglycerol (TG) and total cholesterol (TC) quantification kits were purchased from Abcam (Cambridge, MA, USA). Taqman^®^ Universal Mastermix, Taqman^®^ probes and the high capacity RNA-to-cDNA kit were purchased from Applied Biosystems (Foster City, CA, USA).

### 3.2. Preparation of CP

The CP was prepared using the solvent evaporation method as previously described [15]. Briefly, chrysin and EPL were dissolved in tetrahydrofuran at a molar ratio of 1:3, placed in a 40 °C water bath, and stirred for 4 h. Empty phytosomes (vehicle) were made by excluding chrysin.

### 3.3. Animals and Diets

Male C57BLKS/J-db/db mice (5 weeks old) were purchased from Central Lab, Animal Inc. (Seoul, Korea). Four mice were housed per cage at 22 ± 2 °C and 50 ± 10% relative humidity, with a 12 h light:dark cycle. After 1 week acclimatization period, the mice were randomly divided into six groups (8 mice in each group): (1) non diabetes group (m+/db, C57/BLK/J m +/db), (2) diabetes control group (Control, C57/BLK/J db/db), (3) positive control group (Metformin, C57/BLK/J db/db + 200 mg metformin/kg), (4) db/db + chrysin group (Chrysin, C57/BLK/J db/db + 100 mg chrysin/kg), (5) db/db + CP (CP, C57/BLK/J db/db + 100 mg chrysin equivalent/kg), and (6) db/db + vehicle group (Vehicle, C57/BLK/J db/db + EPL used for CP preparation). The samples containing the indicated dosage were dispersed in PBS and daily through oral gavage for 9 weeks. The dosage of chrysin to be administered was determined based on a chemopreventive study of chrysin [51]. The chrysin content in CP was quantified using HPLC as previously described [18]. Animals had free access to standard chow diet (SAFE, Villemoison-sur-Orge, France) and tap water. Body weight and food intake were recorded weekly. Permission for the animal experiments was granted by Kookmin University Institutional Animal Care and Use Committee (KMUIACUC-2019-01) and continued for 9 weeks.

### 3.4. Blood Biochemical Analysis

The mice were sacrificed after 15 h of fasting and blood samples were collected. Serum was obtained by centrifuging the blood at 2000× *g* for 15 min at 4 °C. Serum alanine aminotransferase (ALT), aspartate aminotransferase (AST), TG, TC, and high-density lipoprotein cholesterol (HDL-c), were measured using a blood analyzer (FUJI DRI-CHEM 3500i, Fujifilm, Tokyo, Japan). Low-density lipoprotein cholesterol (LDL-c) was calculated by applying the Friedewald equation [52]. HbA1c levels in whole blood were determined using an assay kit (Crystal Chem, Downer Grove, IL, USA). Serum insulin and leptin levels in serum were determined using assay kits (Crystal Chem, Downer Grove, IL, USA) according to the manufacturers’ instructions. Homeostasis model assessment of insulin resistance (HOMA-IR) index was calculated as follows: HOMA-IR = fasting insulin (μU/mL) × fasting blood glucose (mmol/L)/22.5 [53].

### 3.5. Oral Glucose Tolerance Test (OGTT)

OGTT was conducted at week 7 after 12 h fasting using a slightly modified method described by Kim et al. [19]. Blood was collected from the tail veins of mice at 0, 15, 30, 60, 90, and 120 min after oral administration of glucose (1.5 g/kg BW). Blood glucose levels were measured using a blood glucose analyzer (Allmedicus, Seoul, Korea). The area under the curve (AUC) for the experimental groups was calculated as described by Pruessner et al. [54]. 

### 3.6. RNA Extraction and Quantitative Real-Time PCR

Total RNA was extracted from the liver and femoral muscle using QIAzol^®^ lysis reagent (QIAGEN, Hilden, Germany) according to the manufacturer’s instruction. Single-stranded cDNA was synthesized using a cDNA kit (Applied Biosystems) and qRT-PCR was performed as described elsewhere [55]. Relative quantification of target mRNA (glucose-6-phosphatase (G6Pase) phosphoenolpyruvate carboxykinase (PEPCK), and hexokinase 2 (HK2) in the liver; glucose transporter 4 (GLUT4), hexokinase 2 (HK2), peroxisome proliferator-activated receptor γ (PPARγ) in femoral muscle) was performed using the comparative CT method by normalizing to the value of the housekeeping gene β-actin. All experiments were conducted in triplicate.

### 3.7. GLUT4 Translocation on the Plasma Membrane of Femoral Muscle

The level of GLUT4 translocation on the plasma membrane of femoral muscle was analyzed using a plasma membrane extraction kit (Abcam, Cambridge, MA, USA) according to a previously published method [56]. The relative intensity of GLUT4 band was normalized to Na+/K+-ATPase and quantified using Image Lab 5.1 software package (BioRad, Hercules, CA, USA).

### 3.8. Statistical Analysis

All data were presented as mean ± standard error. Statistical analysis was performed using Prism 8 (GraphPad Software, Inc., CA, USA). All data were compared using a one-way ANOVA test. A Fisher’s least significant difference (LSD) multiple comparisons test was used to compare the control and treatment groups. In statistical analysis, a *p*-value of <0.05, <0.01, <0.001 was considered as statistically significant.

## 4. Conclusions

CP supplementation significantly improved FBG levels and glucose tolerance in db/db mice compared with the control group. The insulin levels of CP-treated db/db mice were significantly decreased and the surrogate biomarker for insulin resistance, HOMA-IR, was consistently reduced. CP inhibited gluconeogenesis by down regulation of PEPCK and promoted glucose uptake in the skeletal muscle of db/db mice. The insulin levels of CP-treated db/db mice were significantly decreased and the surrogate biomarker for insulin resistance, especially HOMA-IR level, was lower than the chrysin group. CP inhibited gluconeogenesis by down regulating PEPCK and promoted glucose uptake in the skeletal muscle of db/db mice. CP showed greater antidiabetic performance compared with the control by ameliorating insulin resistance in db/db mice. CP also demonstrated an additional effect not observed with chrysin. The antidiabetic effect of CP was probably due to increased bioavailability of the nano-sized chrysin-loaded phytosome formulation.

## Figures and Tables

**Figure 1 molecules-25-05503-f001:**
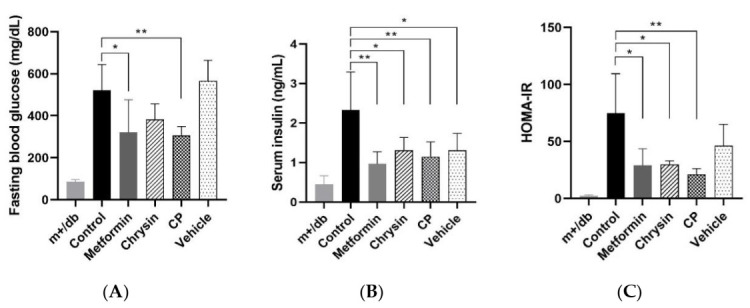
The effects of CP on (**A**) blood glucose, (**B**) serum insulin and (**C**) HOMA-IR. CP, chrysin-loaded phytosomes; HOMA-IR, homeostasis model assessment of insulin resistance. Each value represents mean ± standard error (*n* = 8). * *p* < 0.05, ** *p* < 0.01 compared with control group.

**Figure 2 molecules-25-05503-f002:**
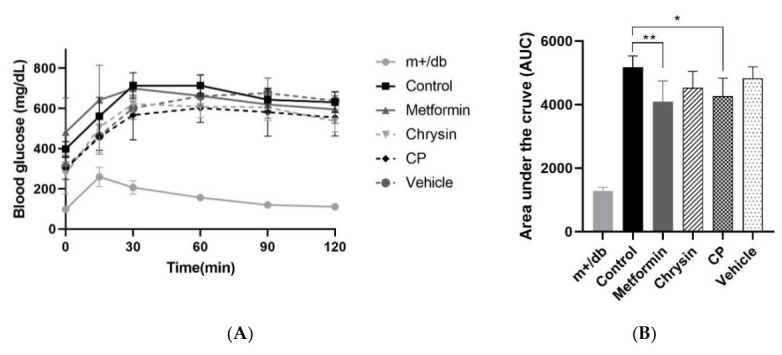
The effects of CP on (**A**) OGTT and its (**B**) AUC. OGTT, oral glucose tolerance test; AUC, area under the curve; CP, chrysin-loaded phytosomes. Each value represents mean ± standard error (*n* = 8). * *p* < 0.05, ** *p* < 0.01 compared with control group.

**Figure 3 molecules-25-05503-f003:**
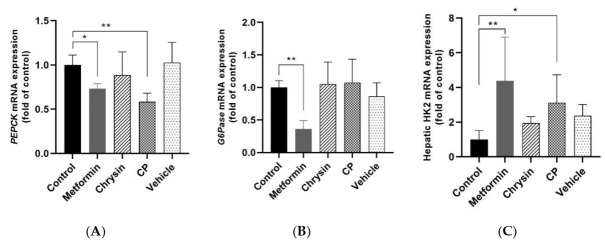
Effects of CP on the glucose metabolism related gene expression in the liver of db/db mice. (**A**) PEPCK, (**B**) G6Pase and (**C**) HK2. PEPCK, phosphoenolpyruvate carboxykinase; G6Pase, glucose-6-phospatase; HK2, hexokinase; CP, chrysin-loaded phytosomes. Each value represents mean ± standard error (*n* = 8). * *p* < 0.05, ** *p* < 0.01 compared with control group.

**Figure 4 molecules-25-05503-f004:**
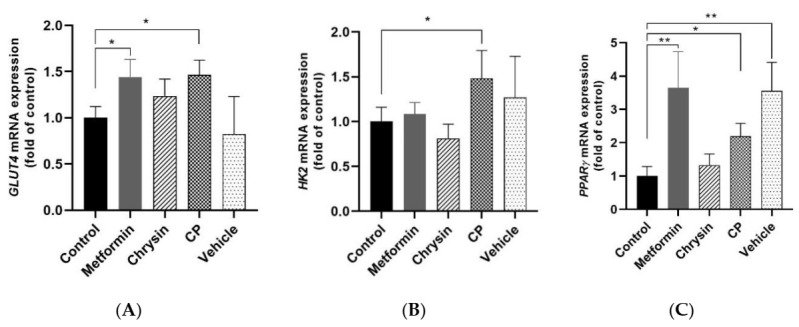
The effects of CP on the glucose uptake related gene expression in the skeletal muscle of db/db mice. (**A**) GLUT4, (**B**) HK2, and (**C**) PPARγ GLUT4, glucose transporter 4; HK2, hexokinase 2; PPARγ, peroxisome proliferator-activated receptor γ; CP, chrysin-loaded phytosomes. Each value represents mean ± standard error (*n* = 8). * *p* < 0.05, ** *p* < 0.01 compared with the control group.

**Figure 5 molecules-25-05503-f005:**
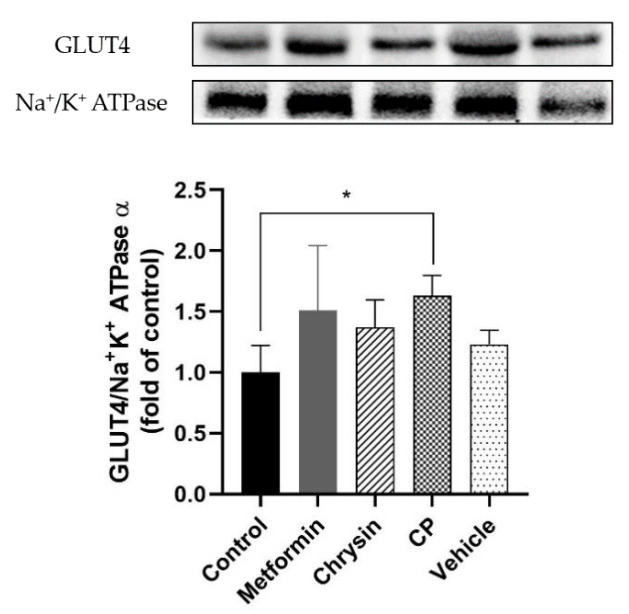
The effects of CP on the GLUT4 plasma translocation in the skeletal muscle of db/db mice. Each value represents mean ± standard error (*n* = 8). * *p* < 0.5 compared to the control group.

**Table 1 molecules-25-05503-t001:** Effect of CP administration on blood biochemical parameters.

Group	m+/db	Control	Metformin	Chrysin	CP	Vehicle
ALT (U/L)	43 ± 14	120 ± 35	85 ± 32	139 ± 38	77 ± 23	88 ± 39
AST (U/L)	87 ± 27	183 ± 53	136 ± 47	193 ± 26	129 ± 26	140 ± 42
TG (mg/dL)	126 ± 16	288 ± 6	257 ± 93	392 ± 83	195 ± 76	140 ± 29 *
TC (mg/dL)	103 ± 11	132 ± 15	131 ± 15	116 ± 14	113 ± 15 *	107 ± 18 *
HDL-c (mg/dL)	80 ± 9	96 ± 7	93 ± 9	85 ± 15	90 ± 14	82 ± 12
LDL-c (mg/dL)	49 ± 8	93 ± 10	90 ± 18	105 ± 18	62 ± 18 *	52 ± 16 **
Leptin (ng/mL)	1.2 ± 0.9	22 ± 7	26 ± 9	23 ± 11	18 ± 5	21 ± 9
HbA1c (%)	4.6 ± 0.2	12.0 ± 2.1	9.0 ± 2.1 *	10.4 ± 2.4	9.8 ± 0.4	9.4 ± 0.6

CP, chrysin-loaded phytosomes; ALT, alanine aminotransferase; AST, aspartate aminotransferase; TG, triacylglycerol; TC, total cholesterol; HDL-c, high-density lipoprotein cholesterol; LDL-c, low-density lipoprotein. Each value represents mean ± standard error (*n* = 8). * *p* < 0.05, ** *p* < 0.01 compared with the control group.

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
