# Peer review of "The Effect of Chrysin-Loaded Phytosomes on Insulin Resistance and Blood Sugar Control in Type 2 Diabetic db/db Mice"

_molecules, 2020, doi:10.3390/molecules25235503_

Round 1

Reviewer 1 Report

Page 1, line 43, you may better explain the effects of chrysin towards renal function. It abrogates renal dysfunction.

Page 2, is chrysin bioavailability also depending on microbiota? are there data about the effects of chrysin bioavailability in relation to the vegetal matrix? if yes, describe these factors.

Page 2, line 104, for the 1st time, HbA1c was evaluated with chrysin. It does not undergo significant variation after chrysin administration. You write HbA1c can probably predict the risk of diabetic  complications better than other glycemic control makers. However, even if the dose of chrysin you used did non alter HbA1c levels, may the other effects counterbalance this aspect? In this case, you may describe these effects

Page 6

Line 207, comma has to be deleted.

Page 7, lines 217-224. The increase of GLUT4 translocation is linked to AMPK stimulation? do you have this information?

Author Response

We thank you for your valuable comments, and have responded to them below.

Page 1, line 43, you may better explain the effects of chrysin towards renal function. It abrogates renal dysfunction.

Response:

Accordingly, we have elaborated the sentence as follows (The revised manuscript (L.46-49):

Chrysin treatment for 16 weeks significantly reduced renal inflammation by the suppression of renal TNF-a production and NF-kB nuclear translocation in high-fat diet and streptozotocin-induced type 2 DM rats. The reduction of inflammation and oxidative stress due to chrysin treatment resulted in the restoration of renal function.

Page 2, is chrysin bioavailability also depending on microbiota? are there data about the effects of chrysin bioavailability in relation to the vegetal matrix? if yes, describe these factors.

Response:

As indicated in L. 57-64, the bioavailability of chrysin is extremely low due to poor absorption, and rapid systematic elimination. The very limited studies available suggest that the effect of gut microbiota on the availability of chrysin may not be significant.

The following sentences were added to the revised manuscript (L. 62-65 and L. 70-72):

Regarding the effect of chrysin on the intestinal environment and microbiome, the administration of chrysin (100 mg/kg, 18 wks) decreased high fructose-mediated upregulated GLUT5 mRNA expression but did not improve dysbiosis in rats [14].

Rani et al. [17] has reported that the phytosome of Casuarina equisetifolia extract containing chrysin improved antidiabetic and antihyperlipidemic activities in Wistar rats.

Page 2, line 104, for the 1st time, HbA1c was evaluated with chrysin. It does not undergo significant variation after chrysin administration. You write HbA1c can probably predict the risk of diabetic complications better than other glycemic control makers. However, even if the dose of chrysin you used did not alter HbA1c levels, may the other effects counterbalance this aspect? In this case, you may describe these effects.

Response:

Although we do not know exact reason for the lack of a statistically significant difference in HbA1c between the control and CP groups but it might be ascribed to the insufficient dosage or the high biological variation in HbA1c. Other possible interferences such as hemoglobinopathies, genetic defects and the presence of other illnesses also might have affected the HbA1c level (Gallagher et al., 2009)

Gallagher, E. J.; Le Roith, D.; Bloomgarden, Z. Review of hemoglobin HbA1c in the management of diabetes. J. Diabetes 2009, 1, 9-17.

Page 6 Line 207, comma has to be deleted.

Response:

We have deleted the comma (L. 222).

Different animal models and experimental periods might have affect the results, even though the exact reasons for the discrepancy are not clear.

Page 7, lines 217-224. The increase of GLUT4 translocation is linked to AMPK stimulation? do you have this information?

Response:

The objective of the present study was to confirm the antidiabetic effect of CP in type 2 DM db/db mice as a continuation of our previous study, in which we demonstrated that CP promoted glucose uptake in C2C12 skeletal muscles through GLUT 4 and PPARg translocation. We did not examine the signaling pathways promoting glucose uptake in the skeletal muscles. However, AMPK-mediated glucose promotion might be involved in improved glucose uptake by CP because chrysin treatment activated AMPK and induced brown fat formation in 3T3-L1 adipocytes (Choi and Yun, 2016).

Choi, J. H.; Yun, J. W. Chrysin induced brown fat-like phenotype and enhances lipid metabolism in 3T3-L1 adipocytes. Nutrition, 2016, 32, 1002-1010.

Reviewer 2 Report

Before publication, please explain the following points.

(1) Please comment on the animal’s well-being, did you monitor them for

   stress/discomfort within the cage?

(2) In this study, the authors believe that the poor solubility and bioavailability of chrysin limit their practical application. Please explain the basis of the proportion of CP prepared in this experiment? Is there any evidence of increased solubility in vitro?

(3) The authors describe that CP shows greater antidiabetic performance by inhibiting

gluconeogenesis and promoting glucose uptake in db/db mice. Please provide more information about its pharmacological evidence and mechanism.

Author Response

We thank you for your valuable comments, and have responded to them below

Before publication, please explain the following points. 

(1) Please comment on the animal’s well-being, did you monitor them for stress/discomfort within the cage?

Response:

The animal experiment was conducted according to the guidelines of Kookmin University Institutional Animal Care and Use Committee (KMUIACUC-2019-01). The animals were monitored twice a day for any incidence of the abnormal appearance of hair condition, eyes, and nose or behavioral posture and movement; we also examined them for injury and diseases and addressed any incidence as necessary. The cages were cleaned twice a week, and the litter was replaced at each cleaning.

(2) In this study, the authors believe that the poor solubility and bioavailability of chrysin limit their practical application. Please explain the basis of the proportion of CP prepared in this experiment? Is there any evidence of increased solubility in vitro?

Response:

In our previous study, the solubility of CP was improved by 60-fold compared with that of free chrysin. The cumulative release characteristics of chrysin determined by the dialysis bag method indicated that CP displayed continuous release even after 24 h (43% release). Contrarily, free chrysin release plateaued at 6 h with no further release (22% release).

Different phospholipid matrices resulted in significant differences in size, mechanical properties, and CP solubility. The most stable CP was obtained with EPL at a molar ratio of 1:3 (chrysin: EPL, CEP-1:3). The average size of CEP-1:3 was 117 nm with uniform size distribution (polydispersity index: 0.30; zeta potential of -31 mV (Kim et al., 2019)

Kim, S.-M.; Jung, J.-I.; Chai, C.; Imm, J.-Y. Characteristics and glucose uptake promoting effect of chrysin-loaded phytosomes prepared with different phospholipid matrices. Nutrients 2019, 11, 2549.

(3) The authors describe that CP shows greater antidiabetic performance by inhibiting gluconeogenesis and promoting glucose uptake in db/db mice. Please provide more information about its pharmacological evidence and mechanism.

Response:

The following sentences have been added to the revised manuscript (L. 158-162):

Type 2 DM patients typically display increased hepatic gluconeogenesis. Metformin, one of the most effective drugs in treating type 2 DM, significantly reduces hepatic gluconeogenesis without increasing insulin secretion [33]. Thus, the gene expressions of two gluconeogenesis rate-limiting enzymes, G6Pase and PEPCK, are effective molecular targets for treating type 2 DM [34].

Additionally, metformin restored insulin-stimulated glucose transport in insulin-resistant human skeletal muscle (Galuska et al., 1991). Pioglitazone is an antidiabetic drug used to treat type 2 DM. It selectively stimulated PPARg and reduced insulin resistance by modulating gene expressions in adipocytes and other tissues, such as skeletal muscle (Stumvoll and Haring, 2002). GLUT4 translocation from cytoplasm to the plasma membrane is the final metabolic response to insulin signal transduction, and the impairment of GLUT4 signaling and translocation leads to insulin resistance in skeletal muscle (Bouzakri et al., 2005).

Galuska, D.; Zierath, J.; Thorne, A.; Sonnenfeld. T.; Wallberg-Henriksson, H. Metformin increases insulin-stimulated glucose transport in insulin-resistant human skeletal muscle. Diabete Metab.1991, 17,159-63.

Bouzakri, K.; Koistinen, H. A.; Zierath, J. R. Molecular mechanisms of skeletal muscle insulin resistance in type 2 diabetes. Curr. Diabetes Rev. 2005, 1, 167-174.

Similar explanations have been included in the revised manuscript (L. 224-231)

Reviewer 3 Report

Seong-min Kim et al. reviewed and analyzed the anti-diabetic properties of chrysin-loaded phytosome formulation in diabetic mice. The manuscript is focused and accurately referenced. Basically, it's well written and consistent with the objectives of the “Molecules” Journal, but there are some major flaws that need to be addressed. Suggested revisions are included below. These comments are intended as a way of enhancing the consistency and readability of the text.

Minor comments:

  • Line 9: Please change “incluing” with “including”
  • Line 15: Please remove one of the two dots
  • Lines 16-19: “The principal biomarkers of insulin resistance, such as oral glucose tolerance test and homeostatic model assessment for insulin resistance, were significantly improved in the CP group (p< 0.05) while hemoglobin A1c resulted in improving trend (0.05 < p< 0.1)”. What does this mean? I don't think we may consider oral glucose tolerance test (OGTT) as a biomarker for insulin resistance (see reference doi: 10.1097/COH.0b013e32833ed177). I, therefore, invite the authors to check and correct this assertion where it has been written in the text.
  • Lines 30-32: “Diabetes mellitus (DM) is a metabolic disorder and characterized by chronic hyperglycemia”. This sentence should be written as “Diabetes mellitus (DM) is a metabolic disorder characterized by chronic hyperglycemia”.
  • Please describe better type 2 diabetes in the "Introduction" section, as this article focuses specifically on this metabolic disorder.
  • Lines 43-45: “Chrysin treatment for   16   weeks significantly enhanced diabetic nephropathy through suppression of tumour necrosis factor-α expression regarding antidiabetic related activities [6]”. What does this sentence mean? What does it mean that treatment with Chrysin significantly enhanced diabetic nephropathy? Please explain and revise it.
  • In all the studies referred to in the "Introduction" and "Results and Discussion" sections, chrysin is administered at different times than the nine weeks chosen by the authors. But why did the authors choose to treat their db/db mice for nine weeks?
  • It is important to better describe the experimental groups examined before discussing the findings in Table 1. Therefore, lines 244-249 should also be mentioned in paragraph 2.1 for a better understanding of the findings shown and described.
  • Please indicate in the text what the abbreviations are referring to. Above all, since there is no list of abbreviations in the manuscript.
  • “Type 2 diabetes” should always be written in the same way throughout the text. Authors are therefore encouraged to review and correct where written differently.
  • Line 135: “These results suggest that chrysin bioavailability was increased by the complexation with EPL”. What are the findings you are referring to? What are the findings that will enable you to affirm this? Please clarify and justify the statement.

Major comments:

  • How do the authors explain the effects (in some cases better) that the vehicle exerts on certain blood biochemical parameters, such as LDL, TG, and TC, compared to CP?
  • In order to really and better appreciate the effects of CP administration on blood biochemical parameters, a comparison should be made not only with the Control group but also with the Chrysin group.
  • The authors often stated in the text that CP promoted glucose utilization in the skeletal muscle by triggering glucose uptake-related gene expression in the skeletal muscle. However, this assertion should be verified by the authors by performing a glucose uptake assay.
  • Insulin resistance in muscle and adipose tissue plays a key role in whole-body insulin resistance. Furthermore, since Glut4, Hk2 and Pparγ (in particular the isoform Pparγ2) are expressed primarily in skeletal muscle and adipose tissue, I believe that the authors should evaluate the effects of CP on the glucose uptake (in terms of gene/protein expression, Glut4 translocation, and glucose uptake assay) also in the adipose tissue of db/db-treated mice.
  • Lines 303-304: “The antidiabetic effect of CP was probably due to increased bioavailability of the nano-sized chrysin-loaded phytosome formulation”. How did the authors come to this conclusion? What are the experiments performed and the results shown in the manuscript, which allow the authors to justify this conclusion?

Author Response

We thank you for your valuable comments, and have responded to them below.

Seong-min Kim et al. reviewed and analyzed the anti-diabetic properties of chrysin-loaded phytosome formulation in diabetic mice. The manuscript is focused and accurately referenced. Basically, it's well written and consistent with the objectives of the “Molecules” Journal, but there are some major flaws that need to be addressed. Suggested revisions are included below. These comments are intended as a way of enhancing the consistency and readability of the text.

Minor comments:

Line 9: Please change “incluing” with “including”

Response:

We have corrected the spelling error (L. 9).

Although a various beneficial health effects of natural flavonoids including chrysin have been suggested, poor solubility and bioavailability limit their practical use.

Line 15: Please remove one of the two dots

Response:

As per your suggestion, we have removed a dot (L. 15).

EPL diet (vehicle, the same quantity of EPL was used for CP preparation) for 9 wks.

Lines 16-19: “The principal biomarkers of insulin resistance, such as oral glucose tolerance test and homeostatic model assessment for insulin resistance, were significantly improved in the CP group (p< 0.05) while hemoglobin A1c resulted in improving trend (0.05 < p< 0.1)”. What does this mean? I don't think we may consider oral glucose tolerance test (OGTT) as a biomarker for insulin resistance (see reference doi: 10.1097/COH.0b013e32833ed177). I, therefore, invite the authors to check and correct this assertion where it has been written in the text.

Response:

We have deleted the reporting trends from the text, and sentences in question have been revised for clarification (L.16-18, L. 112-113, and L. 139-141 in the revised manuscript):

The principal biomarkers of insulin resistance, such as the oral glucose tolerance test and homeostatic model assessment of insulin resistance, were significantly improved in the CP group (p < 0.05). (L.16-18).

The level of glycated hemoglobin (HbA1c) in the metformin group decreased significantly (p < 0.05). (L. 112-113)

The metformin group exhibited consistent improvements in all 3 biomarkers, while the CP group displayed significant differences in both FBG and HOMA-IR. (L. 139-141)

As you pointed out in the referenced article (doi: 10.1097/COH.0b013e32833ed177), biomarkers are considered as surrogate endpoints and should have clinical relevance. In general, fasting blood glucose and insulin values obtained from the homeostatic model assessment (HOMA) are used as a surrogate insulin resistance index. Additionally, the oral glucose tolerance test (OGTT) has been used for a more comprehensive assessment of glucose kinetics in the body (Kim, 2014).

Kim, M. K. Clinical characteristics of OGTT-derived hepatic-and muscle insulin resistance in healthy young men. J. Exerc. Nutrition Biochem. 2014, 18, 385-392.

Shan, X.; Wang, X.; Jiang, H.; Cai, C.; Hao, J.;Yu, G. Fucoidan from ascophyllum nodosum suppresses postprandial hyperglycemia by inhibiting Na+/glucose cotransporter 1 activity. Mar. Drugs 2020, 18, 485.

Nonaka, Y.; Takeda, R.; Kano, Y.; Hoshino, D. Effects of acute 3-h swimming exercise on insulin secretion capacity of pancreatic islets. Japanese J. Phys. Fit. Sports Med. 2020, 9, 173-179.

Lines 30-32: “Diabetes mellitus (DM) is a metabolic disorder and characterized by chronic hyperglycemia”. This sentence should be written as “Diabetes mellitus (DM) is a metabolic disorder characterized by chronic hyperglycemia”.

Response:

According to your suggestion, we have changed the sentence as follows (L. 30 in the revised manuscript)

Diabetes mellitus (DM) is a metabolic disorder characterized by chronic hyperglycemia.

Please describe better type 2 diabetes in the "Introduction" section, as this article focuses specifically on this metabolic disorder.

Response:

According to your suggestion, we have added following sentences (L. 34-37 in the revised manuscript):

Genetic predisposition, obesity, and lack of exercise are major risk factors for of type 2 DM. Pancreatic b-cell increase insulin production for glucose disposal in the early stage of insulin resistance; however, extended hyperglycemic conditions eventually lead to type 2 DM [3].

Lines 43-45: “Chrysin treatment for 16 weeks significantly enhanced diabetic nephropathy through suppression of tumour necrosis factor-α expression regarding antidiabetic related activities [6]”. What does this sentence mean? What does it mean that treatment with Chrysin significantly enhanced diabetic nephropathy? Please explain and revise it.

Response:

The sentence in question has been was revised for clarification (L.46-49 in the revised manuscript):

Chrysin treatment for 16 weeks significantly reduced renal inflammation by suppression of renal TNF-a production and NF-kB nuclear translocation in high fat diet and streptozotosin-induced type 2 DM rats. The reduction of inflammation and oxidative stress due to chrysin treatment resulted in the restoration of renal function.

In all the studies referred to in the "Introduction" and "Results and Discussion" sections, chrysin is administered at different times than the nine weeks chosen by the authors. But why did the authors choose to treat their db/db mice for nine weeks?

Response:

Generally 8-10 weeks of an animal feeding trial is conducted to evaluate effect of the test substance on chronic disease such as DM.

Kim, H. K.; Jeong, J.; Kang, E. Y.; Go, G. Red pepper (Capsicum annuum L.) seed extract improves glycemic control by inhibiting hepatic gluconeogenesis via phosphorylation of FOXO1 and AMPK in obese diabetic db/db Mice. Nutrients 2020, 12(9), 2546 (8 weeks trial)

Ortsäter, H.; Grankvist, N.; Wolfram, S.; Kuehn, N.; Sjöholm, Å. Diet supplementation with green tea extract epigallocatechin gallate prevents progression to glucose intolerance in db/db mice. Nutr. Metab. 2012, 9, 11. (10 weeks trial)

It is important to better describe the experimental groups examined before discussing the findings in Table 1. Therefore, lines 244-249 should also be mentioned in paragraph 2.1 for a better understanding of the findings shown and described.

Response:

According to your suggestion, we have added following sentences (L. 82-85 in the revised manuscript):

In our 9 week feeding trial, the animals were randomly divided into six groups: 1) m+/db, 2) db/db control group (control), 3) positive control group (metformin), 4) db/db + chrysin group (chrysin), 5) db/db + CP (CP), and 6) db/db + vehicle group (vehicle).

Please indicate in the text what the abbreviations are referring to. Above all, since there is no list of abbreviations in the manuscript.

Response:

All abbreviations were defined and added after the conclusion

List of Abbreviations

HbA1c: Hemoglobin A1c

TG: Triacylglycerol

TC: Total cholesterol

FGB: Fasting blood glucose level

ALT: Serum alanine aminotransferase

AST: Aspartate aminotransferase

HDL-c: High-density lipoprotein cholesterol

LDL-c: Low-density lipoprotein cholesterol

AUC: Area under the curve

OGTT: Oral glucose tolerance test

G6 Pase: Glucose-6-phosphatase

PEPCK: Phosphoenolpyruvate carboxykinase

HK2: Hexokinase 2

GLUT4: Glucose transporter 4

PPAR γ: peroxisome proliferator-activated receptor γ

“Type 2 diabetes” should always be written in the same way throughout the text. Authors are therefore encouraged to review and correct where written differently.

Response:

According to your suggestion, we have changed “type 2 diabetes” to “type 2 DM” throughout the text.

Line 135: “These results suggest that chrysin bioavailability was increased by the complexation with EPL”. What are the findings you are referring to? What are the findings that will enable you to affirm this? Please clarify and justify the statement.

Response:

In our previous study [18], CP solubility was improved by 60-fold compared with free chrysin. The cumulated in vitro release characteristics of chrysin indicated that CP displayed continuous release even after 24 h (43% release). Contrarily, free chrysin plateaued at 6 h with no further release (22% release).

Additionally, the following sentences were included in the text (L. 149-153 in the revised manuscript):

Similarly, another study found that silybin, an active flavonoid in milk thistle, increased over 100-fold in plasma when administered as a nanophytosome. Furthermore, pharmacodynamic analysis of rats showed that silybin plasma concentrations increased 1.6 times when administered as a phytosome compared with a free silybin mixture [29]. Orally administered quercetin phytosome also improved plasma quercetin absorption in healthy volunteers (18−50 years) up to 20 times [30].

Major comments:

How do the authors explain the effects (in some cases better) that the vehicle exerts on certain blood biochemical parameters, such as LDL, TG, and TC, compared to CP?

Response:

The vehicle treatment (egg phospholipid) seemed to have had a greater positive effect on blood lipid biochemical parameters such as TG, TC, and LDL-c than on CP, but there was no significant difference between CP and the vehicle treatment for any of the three biomarkers. In the case of biomarkers related to insulin resistance, such as fasting blood glucose, HOMA-IR, and OGTT, only CP displayed improved effects.

Similar to our result, Lee et al. (2014) reported that the administration of phosphatidylcholine improved hyperlipidemia by reducing TG and TC levels in a high-fat, diet-induced mice obesity model. However, a detailed explanation for TG reduction in the vehicle group was not provided.

Lee, H.S.; Nam, Y.; Chung, Y.H.; Kim, H.R.; Park, E.S.; Chung, S.J.; Kim, J.Y., Sohn, U.D.; Kim, H.C.; Oh, K.W.; Jeong, J.H. Beneficial effects of phosphatidylcholine on high-fat diet-induced obesity, hyperlipidemia and fatty liver in mice. Life Sci. 2014, 118, 7-14.

In order to really and better appreciate the effects of CP administration on blood biochemical parameters, a comparison should be made not only with the Control group but also with the Chrysin group.

Response:

Comparisons were made to examine significant differences in blood biochemical parameters among the chrysin, CP, and vehicle treatments, and we found the following: the ALT concentration was significantly lower (p < 0.05) in the CP compared to the chrysin group; the TG and LDL-c levels were significantly lower in the CP and vehicle groups than the chrysin group (p < 0.01); and the CP treatment exhibited greater positive effects on insulin resistance biomarkers such as fasting blood glucose and HOMA-IR compared to the chrysin or vehicle treatments (p < 0.05). Therefore, CP showed greater antidiabetic effects in type 2 DM db/db mice than the chrysin or vehicle treatments.

The authors often stated in the text that CP promoted glucose utilization in the skeletal muscle by triggering glucose uptake-related gene expression in the skeletal muscle. However, this assertion should be verified by the authors by performing a glucose uptake assay.

Response:

In our previous study [18], the effects of the chrysin, CP, and vehicle treatments on glucose uptake were compared using C2C12 skeletal muscle cells. The intensity of cellular 2-NBDG, a fluorescence glucose analog, was quantified after treatment. CP resulted in a significantly higher glucose uptake promoting effect than free chrysin or vehicle in skeletal muscle via stimulating gene expressions of PPARg and GLUT 4. This consistent trend was observed in type 2 DM db/db mice.

Insulin resistance in muscle and adipose tissue plays a key role in whole-body insulin resistance. Furthermore, since Glut4, Hk2 and PPARγ (in particular the isoform Pparγ2) are expressed primarily in skeletal muscle and adipose tissue, I believe that the authors should evaluate the effects of CP on the glucose uptake (in terms of gene/protein expression, Glut4 translocation, and glucose uptake assay) also in the adipose tissue of db/db-treated mice.

Response:

Skeletal muscle is the primary tissue responsible for 80%−90% of glucose disposal and a major insulin resistance site in type 2 DM patients. Thus, we think that the modulation of hepatic gluconeogenesis and the promotion of glucose uptake in skeletal muscle in db/db mice can be used to evaluate the efficacy of CP over free chrysin. As the reviewer pointed out, PPARγ regulates the expression of specific genes in adipose and other tissues, resulting in the net improvement of insulin sensitivity in muscle and liver tissue.

Lines 303-304: “The antidiabetic effect of CP was probably due to increased bioavailability of the nano-sized chrysin-loaded phytosome formulation”. How did the authors come to this conclusion? What are the experiments performed and the results shown in the manuscript, which allow the authors to justify this conclusion?

Response:

In our previous study [18], we characterized a nano-sized CP using a zeta potential analyzer, scanning electron microscopy, X-ray diffraction, and FTIR. The average particle size of the CP was approximately 117 nm with uniform particle size distribution (PDI 0.29). CP solubility was improved 60-fold compared with free chrysin. The cumulative in vitro release characteristics of chrysin indicated that CP displayed continuous release even after 24 h (43% release) while free chrysin plateaued at 6 h with no further release (22% release). The improved bioavailability of phytosomes has been demonstrated in numerous studies.

Chi, C.; Zhang, C.; Liu, Y.; Nie, H.; Zhou, J.; Ding, Y. Phytosome-nanosuspensions for silybin-phospholipid complex with increased bioavailability and hepatoprotection efficacy. Eur. J. Pharm. Sci. 2020144, 105212.

Riva, A.; Ronchi, M.; Petrangolini, G.; Bosiso, S.; Allegrini, P. Improved oral absorption of quercetin phytosomeÒ, a new delivery system based on food grade lecithin. Eur. J. Drug. Metab. Ph. 201944, 169-177.

Telange, D.R.; Patil, A.T.; Pethe, A.M.; Fegade, H.; Anand, S.; Dave, V.S. Formulation and characterization of an apigenin-phospholipid phytosome (APLC) for improved solubility, in vivo bioavailability, and antioxidant potential. Eur. J. Pharma. Sci. 2017, 108, 36–49.

Round 2

Reviewer 3 Report

I don't have any suggestions for the authors. 

Author Response

"The  principal  biomarkers  of  insulin  resistance,  such  as the oral  glucose  tolerance  test  and homeostatic  model  assessment of insulin  resistance,  were  significantly  improved  in  the  CP group (p < 0.05). (L.16-18)"

These are tests for insulin resistance, not biomarkers. Can the authors please correct this or respond?

Response

According to your suggestion, we have changed the sentence as follows (L. 16-17 in the revised manuscript)

Oral glucose tolerance test and homeostatic model assessment for insulin resistance, were significantly improved in the CP group (p < 0.05).
